# Atrial Fibrillation as a Geriatric Syndrome: Why Are Frailty and Disability Often Confused? A Geriatric Perspective from the New Guidelines

**DOI:** 10.3390/ijerph22020179

**Published:** 2025-01-28

**Authors:** Crescenzo Testa, Marco Salvi, Irene Zucchini, Chiara Cattabiani, Francesco Giallauria, Laura Petraglia, Dario Leosco, Fulvio Lauretani, Marcello Maggio

**Affiliations:** 1Geriatric Clinic Unit, University Hospital of Parma, Via Gramsci 14, 43126 Parma, Italy; crescenzo.testa@unipr.it (C.T.); irene.zucchini@unipr.it (I.Z.); chiarac2004@libero.it (C.C.); marcellogiuseppe.maggio@unipr.it (M.M.); 2Department of Medicine and Surgery, University of Parma, Via Gramsci 14, 43126 Parma, Italy; 3Department of Translational Medical Sciences, “Federico II” University of Naples, Via S. Pansini 5, 80131 Naples, Italy; francesco.giallauria@unina.it (F.G.); laura.petraglia@unina.it (L.P.); dario.leosco@unina.it (D.L.)

**Keywords:** frailty, disability, concepts discrepancy, atrial fibrillation, older persons

## Abstract

Atrial Fibrillation can be considered a geriatric syndrome for its prevalence and incidence, its impact on patients’ quality of life, and Health Systems’ economy. The European Society of Cardiology 2024 guidelines introduce a recommendation for maintaining vitamin K antagonist therapy over switching to direct oral anticoagulants in clinically stable elderly patients with atrial fibrillation. This article explores the implications of this indication for the geriatric clinical context. The focus will also be devoted to the need for the stratification of older patients with atrial fibrillation, making an appropriate distinction between frailty and disability.

## 1. Introduction

Atrial fibrillation (AF) has become a prototypical example of a geriatric syndrome due to its increasing prevalence in aging populations and its multifaceted nature. Unlike single-disease models, geriatric syndromes reflect the interplay of multiple comorbidities, functional impairments, and environmental factors.

Despite the lack of generally recognized criteria for defining a geriatric syndrome [1], there are some common features in the current definitions: the higher prevalence with age, the presence of comorbidities and polypharmacy, the impact on quality of life, and its multifactorial nature. Atrial fibrillation (AF) can be fully described if we consider the following categories:

**Prevalence with age**: AF is extremely common in elderly patients, and its incidence significantly increases with age, as with frailty and cognitive decline.

**Comorbidities and Polypharmacy**: In elderly patients, AF is often associated with multiple chronic diseases and a wide range of medications, as a typical feature of geriatric syndromes [2].

**Impact on Quality of Life**: AF has significant effects on mobility, fall risk, and overall well-being, contributing to increased frailty and disability [3].

**Multifactorial Nature**: like many geriatric syndromes, AF results from multiple and complex factors, including not only traditional cardiovascular causes but also chronic inflammation, oxidative stress, and autonomic dysfunction [4].

An extensive amount of the literature supports this approach. Recent epidemiological data have highlighted how the percentage of patients affected by atrial fibrillation or atrial flutter has increased by 137% in the last thirty years. The most affected age groups are the geriatric populations, as the highest prevalence is evident in the age group from 70 to 79 years, whereas the highest incidence is in the patients aging from 65 to 74 years [5]. The older age is typically characterized by the interplay between different diseases and poor functional status. A meta-analysis including over a million patients has shown that the prevalence of frailty in patients with atrial fibrillation is very high, approximately 40%. In these patients, the risk of polypharmacy is remarkable [6,7]. These aspects are closely linked and very often have a negative impact on the quality of patients’ lives [8].

This demographic shift emphasizes the urgency of developing tailored interventions that address both AF and its associated comorbidities. Moreover, recent guidelines emphasize the importance of dynamic assessment tools, such as frailty indices, to guide treatment decisions. From this perspective, the guidelines for the management of atrial fibrillation promote AF-CARE: an approach that emphasizes the need for a multidisciplinary team that includes geriatricians in caring for AF patients. This approach is vital, considering that AF has become a predominant geriatric condition with a considerable burden in terms of morbidity and healthcare resource utilization [9].

The following acronym CARE underlines the importance of a multidisciplinary approach to the patient:**C**: Comorbidity and risk factor management;**A**: Avoid stroke and thromboembolism;**R**: Reduce symptoms by rate and rhythm control;**E**: Evaluation and dynamic reassessment.

### 1.1. Comorbidity and Risk Factor Management

Atrial fibrillation is a complex condition that highlights the intricate interplay between aging, multimorbidity, and polypharmacy, particularly in geriatric populations. Managing comorbidities and associated risk factors in older adults with AF demands a patient-centered approach tailored to their unique vulnerabilities and physiological reserves.

Frailty is a central and multidimensional concept that encapsulates the vulnerability of aging individuals to external stressors. Defined as a clinical state of increased susceptibility to adverse health outcomes, frailty arises from cumulative declines across physical, cognitive, and psychosocial domains. Unlike disability, which entails a loss of autonomy in daily activities, frailty represents reduced physiological reserves without necessarily compromising independence. Two dominant models conceptualize frailty: the frailty phenotype and the cumulative deficit model. The phenotype approach, as proposed by Fried et al., identifies frailty through five key indicators: unintentional weight loss, exhaustion, weakness, slow walking speed, and reduced physical activity. Conversely, the cumulative deficit model defines frailty as the accumulation of health deficits, such as comorbidities and functional impairments, which can be quantified through indices like the Frailty Index [10].

The Clinical Frailty Scale (CFS), extensively validated in the recent literature, provides a pragmatic tool to assess frailty in clinical settings [11]. This nine-point scale integrates multiple dimensions of a patient’s health, ranging from robust health to severe frailty. For example, CFS scores of 4 or lower indicate individuals who are vulnerable but retain functional independence, whereas scores of 5 or higher denote varying degrees of frailty or disability with significant clinical implications. Studies have demonstrated that the CFS is a reliable predictor of adverse outcomes in geriatric patients with AF, including higher risks of hospitalization, bleeding complications, and mortality. Incorporating the CFS into clinical workflows enables tailored treatment strategies, such as adjusting anticoagulation regimens to mitigate bleeding risks in frail patients while preserving therapeutic efficacy [10,11].

The interplay between frailty and polypharmacy further complicates the management of older AF patients. Polypharmacy, commonly defined as the concurrent use of five or more medications, is prevalent in over 50% of older adults with AF [12]. While appropriate polypharmacy can address multiple chronic conditions, inappropriate polypharmacy poses significant risks. Systematic reviews have shown that inappropriate polypharmacy is a leading cause of hospitalizations in older adults, often due to adverse drug reactions, medication non-adherence, and drug–drug interactions. A meta-analysis revealed that inappropriate prescribing contributes to up to 25% of hospital admissions in this demographic [13,14]. Common issues include the overuse of medications with high bleeding risk, such as anticoagulants, without proper monitoring and the underuse of essential drugs that could prevent complications.

Medication errors and malpractice in prescribing practices further exacerbate the issue. Studies highlight cases where older patients are prescribed potentially inappropriate medications (PIMs), leading to avoidable adverse outcomes. For instance, the Beers Criteria and STOPP/START guidelines identify medications that should be used with caution in older adults. Adherence to these guidelines has been shown to reduce hospitalizations and improve overall outcomes. However, their underutilization in routine practice often results in suboptimal care. Addressing these gaps requires rigorous medication reviews, deprescription strategies, and enhanced clinician training [15,16].

A holistic and precise approach to managing polypharmacy involves the integration of tools like the Comprehensive Geriatric Assessment (CGA), which evaluates medication appropriateness within the context of a patient’s overall health status. This process ensures that therapeutic regimens are not only effective but also aligned with the patient’s functional and cognitive capacities. Pragmatic clinical trials that include older adults with polypharmacy and multimorbidity are crucial to generating evidence-based guidelines tailored to this population [17].

Holistic management extends beyond anticoagulation and includes strategies to mitigate modifiable risk factors such as hypertension, obesity, and diabetes. Multidisciplinary care models, exemplified by the AF-CARE framework, emphasize collaboration among cardiologists, geriatricians, and primary care providers to address the complex needs of frail patients. These models have been shown to improve clinical outcomes and enhance quality of life by prioritizing interventions like fall prevention, cognitive support, and nutritional optimization [9].

In conclusion, managing polypharmacy in older adults with AF requires a multidimensional and individualized approach. Addressing inappropriate prescribing practices, leveraging tools like the Clinical Frailty Scale and CGA, and designing inclusive clinical trials are essential steps. By aligning interventions with patients’ unique needs, clinicians can optimize outcomes, minimize risks, and uphold the principles of patient-centered care.

### 1.2. Avoid Stroke and Thromboembolism

A new development stands out by analyzing more carefully what is proposed by the new guidelines regarding the decoagulation of elderly patients.

The authors of this guideline state that “maintaining vitamin K antagonists (VKA) treatment rather than switching to a direct oral anticoagulant (DOAC) may be considered in patients aged ≥75 years on clinically stable therapeutic VKA with polypharmacy in order to prevent excess bleeding risk”.

This recommendation is in class IIb, level of evidence B: this means that the usefulness/efficacy of this indication is less well established by evidence/opinion, and the same guidelines recommend that the right words to explain it are “may be considered” in clinical practice. Level of evidence B is because the data supporting the indication are derived from a single randomized clinical trial [9].

As highlighted in the literature, frailty often exacerbates the risks associated with AF, including higher rates of hospitalizations, falls, and medication-related harm [6].

In everyday clinical practice, we often deal with old therapeutic schemes “inherited” by patients over the years. One of the drugs most used by patients for years is usually Warfarin. The indication of the new guidelines regarding the maintenance of the therapy with Warfarin is extremely important in patients aged 75 or over. The misinterpretation of this suggestion could lead to a therapeutic inertia towards NOACs that is not supported by current scientific evidence. A clear trend in the efficacy of NOACs in elderly patients with a good risk profile has been indeed demonstrated several times [18,19]. One of the major pieces of evidence supporting the efficacy of NOAC use in frail patients with atrial fibrillation has been recently published by the European Geriatric Medicine Society (EuGMS)—Cardiovascular Disease Working Group. Moreover, the EUROSAF study demonstrated that NOAC use in frail elderly patients is safe, effective, and associated with lower mortality [20]. The continuation of VKA therapy in patients aged 75 years or older is supported by a single study published in January 2024. The FRAIL-AF is a multicenter, open-label, randomized controlled superiority trial comparing the use of VKA and switches to a NOAC in 1330 patients with a mean age of 83 years. Patients were randomly assigned to switch from an international normalized ratio-guided VKA treatment to a NOAC or to continue VKA treatment [21]. The study clearly demonstrated that switching an international normalized ratio-guided VKA treatment to a NOAC in frail older patients with atrial fibrillation was associated with more bleeding complications compared with continuing VKA treatment, without an associated reduction in thromboembolic complications. To be eligible for the study, patients had to be 75 or older and “frail”. In particular, the tool chosen for the stratification of frailty is the Groningen Frailty Indicator (GFI), with a cut-off score of 3 or higher. The Groningen Frailty Index is a tool for the stratification of frailty composed of 15 items, by which the progressively higher the score, the greater the degree of functional impairment [22]. A careful analysis of the population and the supplementary material of the FRAIL AF show that the functional domains analyzed in the stratification of frailty were the following: mobility, vision and hearing, nutrition, comorbidity, cognition, and psychosocial and physical fitness. The analysis of the patients switching who went from VKA to NOAC identified many patients with a GFI score ≥ 3, which indicates patients with limited self-care capability, confined to bed or chair and about <50% of waking hours”. Patients with a functional score of 4 or higher are indicated as “completely disabled, cannot carry on any self-care, totally confined to bed or chair”. Regardless of the outcome of the study, there was a critical misidentification and stratification of patients.

Additional evidence underscores the importance of stratifying elderly patients appropriately to optimize outcomes. A fundamental aspect of this stratification is the critical distinction between frailty and disability, as these concepts, while often conflated, have distinct clinical implications. Misidentifying disabled patients as frail can lead to inappropriate therapeutic decisions, such as over- or under-anticoagulation. Among DOACs, apixaban and edoxaban have emerged as preferred options in elderly cohorts due to their favorable safety profiles. Apixaban has demonstrated superiority in reducing stroke and systemic embolism without increasing major bleeding risks, even in patients with renal impairment [23]. This aligns with findings from trials such as ARISTOTLE and ENGAGE AF-TIMI 48, which confirmed the efficacy and safety of apixaban and edoxaban, respectively, in older populations with varying degrees of renal dysfunction and frailty [24,25].

Stratification of patients is paramount in clinical practice, as frailty, renal function, and comorbid conditions significantly influence the risk–benefit profile of anticoagulation therapies. For example, frail patients with mild renal impairment may benefit from apixaban due to its minimal renal clearance compared to dabigatran, which is associated with higher gastrointestinal bleeding risks. Similarly, rivaroxaban may be considered in patients requiring once-daily dosing to improve adherence, especially in those with cognitive or logistical barriers to complex regimens [26]. Renal function should be assessed regularly as both over-anticoagulation and under-anticoagulation pose significant risks in this demographic. Recent insights from pragmatic trials like ELDERCARE-AF have highlighted the importance of dose-adjustments in very elderly patients. This study evaluated low-dose edoxaban in patients aged ≥80 years and demonstrated a significant reduction in stroke risk with a manageable bleeding profile, reinforcing the role of tailored anticoagulation strategies in frail populations [27].

Future research must prioritize inclusive trials that incorporate frailty as a key stratification factor to better inform therapeutic decisions.

Figure 1 shows the distinction between frailty and disability. Although there are numerous tools available for the identification and stratification of frailty, the misidentification between frailty and disability commonly occurs in clinical practice. Thus, it becomes very important to underline that the frail patient, regardless of age and degree of functional impairment, is a patient who still maintains autonomy in daily life activities. The disabled patient, on the other hand, has already lost this autonomy. Disability can be identified at physical, cognitive, and multidimensional levels. Regardless of the underlying cause, this condition is an independent risk factor for bleeding and adverse outcomes, especially above the age of 75 [3]. The recommendation of the new guidelines for the management of atrial fibrillation raises significant geriatric concerns, increasing the existing frequent confusion between the concepts of disability and frailty.

Being disabled and over 75 years old very often means being bedridden, at risk of hypo- or hyper-kinetic delirium, and predisposed to dehydration and acute renal failure; all these conditions contraindicate the use of new oral anticoagulant drugs, as well as decoagulation in general. The geriatric clinical world is already very-well prepared to deal with these conditions. These recommendations shall not be generalized in the real world of clinical practice where patients over 75 years of age are prevalent, and it could be difficult to frame their robustness or frailty. It would be even more difficult to assess the benefit of a VKA or a NOAC anticoagulation regimen as well. Recently, the issue of frailty in elderly patients has been addressed in a systematic manner, making very clear the opportunity to avoid the “therapeutic inertia” in the fit of “truly frail” older patients [28].

### 1.3. Reduce Symptoms by Rate and Rhythm Control

Managing atrial fibrillation in frail elderly patients necessitates a tailored approach that considers their unique clinical presentations and vulnerabilities. Unlike younger populations, frail older adults often exhibit atypical symptoms of AF, including sleep disturbances, hypo-kinetic or hyper-kinetic delirium, and generalized functional decline. Moreover, it is not uncommon for AF to be first recognized through its complications, such as stroke or systemic thromboembolism, rather than its classic symptoms like palpitations or dyspnea. This highlights the importance of proactive and comprehensive management strategies that integrate rate and rhythm control [9]. Rate control, primarily achieved through beta-blockers, calcium channel blockers, or digoxin, remains foundational, yet each agent’s risk–benefit profile must be carefully evaluated to minimize adverse outcomes [29]. Rhythm control, including antiarrhythmic drugs and catheter ablation, may offer symptomatic relief and improved quality of life [30,31]. Notably, recent evidence underscores the efficacy of catheter ablation in selected elderly patients, even those with heart failure, demonstrating reduced hospitalizations and enhanced survival [32]. The identification of extra-pulmonary vein triggers has further expanded the therapeutic potential of ablation in refractory cases [33]. For frail patients, therapeutic decisions should prioritize safety and align with individual functional status, emphasizing a multidisciplinary approach. Non-pharmacological measures, such as optimizing nutrition and physical activity and leveraging wearable monitoring technologies, can integrate medical therapies to address the broader health challenges faced by these patients. Recognizing the heterogeneity in symptom expression and leveraging advanced diagnostic and therapeutic tools are essential for achieving optimal outcomes in this complex population.

### 1.4. Evaluation and Dynamic Reassessment

Clinical trials have long been the gold standard for generating evidence-based medical guidelines. However, their generalizability to older adults with frailty and multimorbidity, who constitute a significant portion of the atrial fibrillation (AF) and heart failure (HF) populations, remains limited [33,34]. Frailty intersects with multimorbidity and aging to present unique challenges in healthcare management. The next inclusion of this population in future clinical trials would be critical for advancing evidence-based, patient-centered care. This section deeply explores the importance of inclusive trials, the role of advanced stratification tools, and the benefits of collaborative care models in improving outcomes for frail and multimorbid patients.

Despite growing recognition of the importance of frailty in clinical practice, many trials systematically exclude frail individuals. Studies demonstrate that up to 87% of older adults hospitalized with heart failure or atrial fibrillation are ineligible for participation in major clinical trials due to stringent inclusion criteria, such as age cutoffs, the absence of significant comorbidities, or functional limitations [35,36]. These exclusions have led to a paucity of evidence for treating these populations and hinder clinicians’ ability to make informed, tailored decisions.

The exclusion is often driven by concerns about adverse events that frail individuals are more likely to experience during trials, including medication-related harm and hospitalizations. For instance, trials like EMPEROR and DELIVER recruited participants with relatively lower frailty levels, as assessed by simplified indices that omitted key markers such as cognitive function, polypharmacy, and fall risk [37,38,39,40]. These gaps leave clinicians with data to extrapolate from younger, healthier populations in order to manage older, frailer patients. As potential dangerous consequence the most common clinical approach consists of suboptimal or overly cautious care.

Thus, inclusion of frail and multimorbid populations in clinical trials becomes an ethical imperative rather than just a scientific necessity. Frail individuals represent the real-world complexity of aging populations and account for a significant proportion of healthcare resource utilization. Observational data suggest that frailty confers a higher risk of adverse events from both disease progression and treatment, yet the balance of risks and benefits for various interventions remains unclear without robust trial data [41].

Pragmatic clinical trial designs are a promising solution. Unlike traditional randomized controlled trials (RCTs), pragmatic trials feature broader inclusion criteria and focus on real-world settings. They allow for the evaluation of interventions in diverse patient populations, including those with frailty, polypharmacy, and multimorbidity. Trials like FRAIL-AF, despite their limitations, have paved the way for frail older adults to be included. However, improvements in frailty stratification are necessary to enhance data reliability and applicability.

Frailty is a continuum rather than a binary state, complicating its measurement and classification. Not all existing tools, including Fried Frailty Phenotype, capture all dimensions of frailty. This limited scope risks conflating frailty with disability, as was observed in the FRAIL-AF trial, where patients with significant functional impairments were labeled as frail.

Frailty assessment is a clinical practice that improves outcomes even in cardiovascular settings other than atrial fibrillation and heart failure [42].

Advanced frailty indices offer several advantages:Improved Patient Selection: by distinguishing between frailty and disability, these tools help identify patients who may benefit most from interventions, avoiding the risks of overtreatment or undertreatment [43].Enhanced Predictive Accuracy: multidimensional assessments provide a clearer picture of treatment risks and benefits, improving the reliability of trial outcomes.Alignment with Real-World Practice: these tools mirror the complexity of clinical settings, making trial findings more applicable to everyday patient care.

The integration of multidimensional frailty indices into trials like EUROSAF has demonstrated the feasibility and value of this approach, highlighting the safer and more effective use of NOACs in properly stratified frail populations.

Managing frail and multimorbid patients requires expertise that spans multiple disciplines. Cardiologists excel in treating cardiovascular conditions like AF and HF, but geriatricians bring a broader perspective that addresses frailty, disability, and polypharmacy. Collaborative care models, such as those promoted by the AF-CARE framework, integrate these perspectives to deliver holistic, patient-centered care.

## 2. Benefits of Collaborative Models in Managing Frailty and Multimorbidity

Collaborative models of care offer a transformative approach to managing frail and multimorbid patients, addressing the complexities of their healthcare needs in a way that transcends traditional, single-specialty care. By bringing together cardiologists and geriatricians, these models provide a comprehensive framework for assessment, treatment planning, and resource utilization. Their benefits are far-reaching, improving both clinical outcomes and the overall well-being of vulnerable patient populations.

One of the most significant advantages of collaborative care is the ability to perform detailed and multidimensional patient assessments. Geriatricians are uniquely trained to evaluate frailty, cognitive function, psychosocial factors, and other geriatric syndromes that influence health outcomes. These assessments complement the disease-specific focus of cardiologists, who bring expertise in managing conditions like atrial fibrillation (AF) and heart failure (HF). By combining these perspectives, collaborative teams can gain a holistic understanding of a patient’s health status, identifying not only the immediate clinical issues but also the underlying vulnerabilities that may impact treatment success [44]. For example, assessing cognitive function is critical for understanding a patient’s ability to adhere to complex medication regimens or recognize symptoms that require medical attention. Similarly, psychosocial evaluations can reveal barriers to care, such as social isolation or financial difficulties, that may otherwise go unnoticed in a traditional cardiology setting [45].

Personalized treatment planning is another hallmark of collaborative models. By integrating insights from both specialties, care teams can develop regimens that align with the patient’s overall health goals, functional status, and individual preferences. For instance, a cardiologist may recommend a specific anticoagulant therapy for AF, while the geriatrician assesses whether the patient’s frailty, fall risk, or polypharmacy might necessitate adjustments to the standard treatment protocol. This interdisciplinary approach ensures that therapies are not only clinically effective but also feasible and safe within the context of the patient’s broader health profile [46]. Personalization also extends to non-pharmacological interventions, such as physical therapy or nutritional support, which can be tailored to address specific deficits in mobility or nutritional status. By focusing on the whole patient rather than just the disease, collaborative models promote a higher standard of care that is both individualized and patient-centered [47].

Resource optimization is another critical benefit of collaborative care. Multidisciplinary teams could reduce hospitalizations, minimize medication-related adverse events, and improve overall adherence to treatment plans [48]. For frail and multimorbid patients, frequent hospital admissions often signal gaps in outpatient management or unaddressed social determinants of health. By proactively addressing these issues through comprehensive assessments and personalized plans, collaborative teams can help patients maintain stability and avoid unnecessary acute care. For example, a geriatrician might identify early signs of functional decline that could predispose a patient to falls or infections, enabling timely interventions that prevent hospitalizations. Likewise, cardiologists can leverage their expertise to fine-tune cardiovascular therapies, reducing the risk of decompensation or adverse drug interactions.

Another way of collaborative models optimizing resources is the efficient use of healthcare personnel and infrastructure. By integrating specialists into a cohesive team, these models reduce duplication of services and streamline care delivery. For instance, a shared electronic health record system can facilitate communication among team members, ensuring that all providers have access to the same comprehensive patient data [47,48]. This reduces the likelihood of conflicting recommendations and allows for coordinated follow-up care. Moreover, multidisciplinary case conferences enable providers to discuss complex cases, pooling their expertise to arrive at the best possible care plan. These efficiencies not only improve outcomes but also reduce the burden on healthcare systems, which are increasingly strained by the demands of an aging population.

The benefits of collaborative care extend beyond clinical outcomes to encompass improvements in patients’ quality of life. Frail and multimorbid patients often experience significant disruptions to their daily lives due to their health conditions, from reduced mobility to social isolation and psychological distress. By addressing these issues holistically, collaborative teams can help patients regain a sense of control and independence. For example, integrating physical rehabilitation into a treatment plan can improve mobility and reduce the risk of falls, while psychosocial support can help patients navigate the emotional challenges of living with chronic illness [49]. Collaborative models also empower patients through shared decision-making, involving them in every step of their care. This approach fosters a sense of partnership between patients and providers, enhancing satisfaction and adherence to treatment plans.

One illustrative example of the benefits of collaborative care is the management of anticoagulation therapy in frail elderly patients with AF. These patients often face a delicate balance between the risks of thromboembolism and bleeding, requiring nuanced decision-making that takes into account their overall health status. In a collaborative model, the cardiologist provides expertise on the efficacy and safety of various anticoagulants, while the geriatrician assesses factors such as fall risk, cognitive function, and the patient’s ability to adhere to therapy. Together, they can develop a plan that minimizes risks while optimizing therapeutic outcomes. This interdisciplinary approach ensures that the patient’s unique needs are addressed, reducing the likelihood of adverse events and improving long-term outcomes. Furthermore, collaborative models facilitate the integration of innovative care strategies, such as telemedicine and remote monitoring, which are particularly valuable for managing frail and homebound patients. By leveraging technology, teams can provide continuous care and early intervention for issues that might otherwise lead to hospitalizations. For example, remote monitoring of vital signs and weight can help detect early signs of heart failure exacerbation, prompting timely adjustments to therapy [50,51]. Similarly, telemedicine consultations enable patients to access specialized care without the need for frequent travel, which can be physically and emotionally taxing for frail individuals.

Research supports the effectiveness of collaborative models in improving outcomes for frail and multimorbid populations. Studies have shown that integrated care programs reduce hospital readmissions, improve medication adherence, and enhance patients’ overall satisfaction with their care [52]. These benefits are not only clinically significant but also economically advantageous, as they help mitigate the high costs associated with hospitalizations and emergency care. By prioritizing preventive measures and proactive management, collaborative models offer a sustainable approach to meeting the healthcare needs of aging populations.

In conclusion, collaborative care models represent a paradigm shift in the management of frailty and multimorbidity, offering a comprehensive, patient-centered approach that bridges the gap between cardiology and geriatrics. Through detailed assessments, personalized treatment plans, and optimized resource utilization, these models address the unique challenges of caring for vulnerable populations. As the healthcare landscape continues to evolve, the adoption of collaborative models will be essential for improving outcomes and ensuring that all patients receive the high-quality care they deserve.

## 3. Conclusions

The updated guidelines for managing atrial fibrillation (AF) signal a transformative shift toward a multidisciplinary and patient-centered approach, particularly in the context of frail elderly patients. AF, increasingly recognized as a geriatric syndrome, requires nuanced management strategies that account for the complexities of frailty, multimorbidity, and the unique challenges posed by aging populations.

A critical distinction highlighted throughout this review is the differentiation between frailty and disability. Frailty reflects diminished physiological reserves without the loss of autonomy, whereas disability involves a significant decline in functional independence. This distinction is not merely theoretical but has profound clinical implications. Misinterpreting disability as frailty can result in either therapeutic inertia or excessive intervention, both of which carry significant risks. Validated tools such as the Clinical Frailty Scale (CFS) enable precise stratification of patients, guiding interventions that are both effective and individualized.

The practical implications of these insights are multifaceted. First, clinicians must prioritize comprehensive assessments that include frailty, renal function, and cognitive status to inform treatment decisions. Second, the selection of anticoagulants in older adults should favor agents with established safety profiles in this demographic, such as apixaban and edoxaban, which reduce stroke and systemic embolism risks without disproportionately increasing bleeding events. Evidence from trials like ARISTOTLE and ENGAGE AF-TIMI 48 underscores the utility of these agents in frail populations. Furthermore, pragmatic trials, such as ELDERCARE-AF, demonstrate the efficacy of low-dose anticoagulation regimens in reducing stroke risk while managing bleeding complications, emphasizing the value of dose adjustments tailored to individual profiles.

Collaborative care models emerge as a cornerstone of effective management. Integrating expertise from cardiology and geriatrics fosters a holistic approach that addresses not only cardiovascular risks but also the broader determinants of health, such as nutrition, mobility, and psychosocial well-being. Non-pharmacological interventions, including fall prevention programs and wearable monitoring technologies, complement pharmacological strategies and contribute to improved outcomes.

For researchers, this review underscores the urgent need for inclusive trials that incorporate frailty as a stratification criterion. Current evidence is often limited by the exclusion of frail and multimorbid patients, resulting in a knowledge gap that compromises real-world applicability. Future studies must adopt pragmatic designs to evaluate interventions in diverse patient populations, ensuring that findings are both robust and relevant to clinical practice. For clinicians, the message is clear: accurate patient stratification is non-negotiable. Frailty must be assessed using validated indices to avoid conflating it with a disability, thereby ensuring appropriate therapeutic decisions. Furthermore, regular monitoring of renal function and frailty status is critical to dynamically adjust treatments as patients’ conditions evolve.

In conclusion, the updated AF guidelines provide a comprehensive framework for optimizing care in older adults. By integrating multidisciplinary collaboration, precise patient stratification, and evidence-based therapeutic strategies, clinicians can navigate the complexities of managing frailty and multimorbidity. This approach not only aligns with the principles of patient-centered care but also holds the potential to enhance clinical outcomes, improve quality of life, and address the pressing needs of an aging population.

## Figures and Tables

**Figure 1 ijerph-22-00179-f001:**
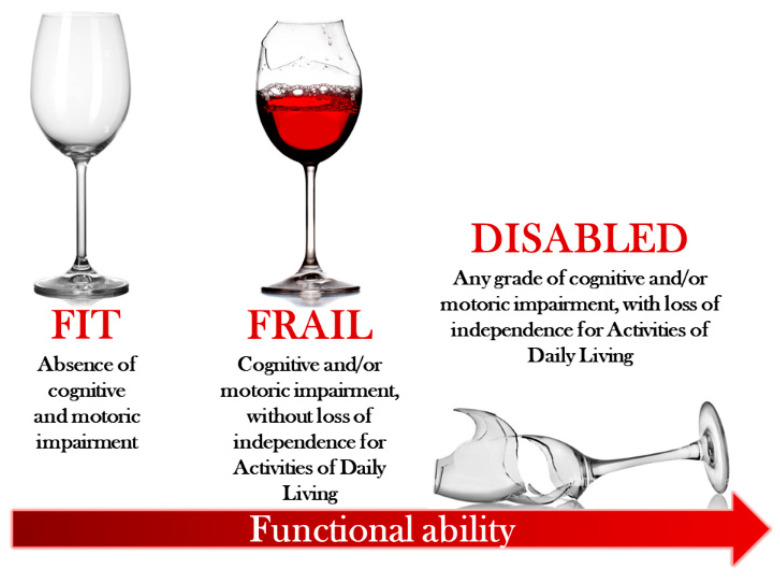
Metaphorical Picture of Fitness, Frailty, and Disability.

## Data Availability

No new data were created or analyzed in this study.

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
