# Peer review of "Atrial Fibrillation as a Geriatric Syndrome: Why Are Frailty and Disability Often Confused? A Geriatric Perspective from the New Guidelines"

_ijerph, 2025, doi:10.3390/ijerph22020179_

Round 1
Reviewer 1 Report
Comments and Suggestions for Authors
I have reviewed the manuscript entitled ‘Atrial Fibrillation as a Geriatric Syndrome: Why are Frailty and Disability often confused? A Geriatric Perspective from the New Guidelines’.
The hypothesis is well-presented however please also mention how can we provide more precise approach in geriatric patients.
Please also mention the necessity of trials including only the geriatric patients.
The potential role of AF ablation in these patients should be evaluated separately in geriatric patients such in heart failure patients. Since the role of AF ablation in heart failure has been demonstrated several times in randomized trials. Please mention this issue citing ‘Comparison of catheter ablation and medical therapy for atrial fibrillation in heart failure patients: A meta-analysis of randomized controlled trials’. Also the role of extra-pulmonary triggers should be noted in these patients citing ‘Catheter Ablation Approaches for the Treatment of Arrhythmia Recurrence in Patients with a Durable Pulmonary Vein Isolation’
Author Response
Reviewer 1 comment:
- “The hypothesis is well-presented, but please also mention how we can provide a more precise approach in geriatric patients.”
- “Please also mention the necessity of trials including only geriatric patients.”
- “The potential role of AF ablation in these patients should be evaluated separately in geriatric patients such as in heart failure patients. Since the role of AF ablation in heart failure has been demonstrated several times in randomized trials, please mention this issue citing ‘Comparison of catheter ablation and medical therapy for atrial fibrillation in heart failure patients: A meta-analysis of randomized controlled trials’. Also, the role of extra-pulmonary triggers should be noted in these patients citing ‘Catheter Ablation Approaches for the Treatment of Arrhythmia Recurrence in Patients with a Durable Pulmonary Vein Isolation.’”
Author Response:
1.More precise approach in geriatric patients
We have expanded our discussion in the “Comorbidity and Risk Factor Management” section to outline more detailed, step-by-step strategies for clinicians assessing older adults with AF. We explicitly include frailty indices (such as the Clinical Frailty Scale) and comprehensive geriatric assessment (CGA) to tailor interventions based on comorbidities, functional status, and cognitive reserve.
2.Necessity of geriatric-focused trials
In the “Evaluation and Dynamic Reassessment” section, we emphasize the importance of designing and conducting clinical trials exclusively or primarily for geriatric patients. We discuss how frail individuals are often excluded from major randomized controlled trials, limiting real-world applicability of the findings. We have included new references and text underlining the ethical and scientific imperative for trials that enroll older adults with multimorbidity and polypharmacy.
3.Role of AF ablation in older and heart failure populations
We have added a dedicated paragraph in “Reduce Symptoms by Rate and Rhythm Control” to address the specific role of catheter ablation in heart failure patients. We now cite and discuss the meta-analysis “Comparison of catheter ablation and medical therapy for atrial fibrillation in heart failure patients: A meta-analysis of randomized controlled trials,” highlighting that selected geriatric patients with preserved functional status could benefit from ablation. Additionally, we acknowledge the importance of extra-pulmonary vein triggers in refractory cases, referencing “Catheter Ablation Approaches for the Treatment of Arrhythmia Recurrence in Patients with a Durable Pulmonary Vein Isolation” to illustrate advanced ablation strategies that can be relevant in older, complex AF patients.
Reviewer 2 Report
Comments and Suggestions for Authors
This manuscript examines atrial fibrillation (AF) as a geriatric syndrome, focusing on its prevalence, complex nature, and profound effects on the elderly population. It emphasizes the distinction between frailty and disability, stressing the need for precise stratification to enhance management strategies. The authors critically assess current guidelines, particularly the recommendation to continue vitamin K antagonist therapy in frail older patients, while advocating for a multidisciplinary approach, such as the AF-CARE framework, to improve patient care. The challenges of applying clinical trial data to older populations are discussed, highlighting the necessity for inclusive and pragmatic research methodologies. The paper concludes by advocating for customized, patient-centered approaches to improve health outcomes and quality of life for elderly individuals with AF.
The discussion on guideline implications would benefit from a more thorough comparison with current clinical practices. Moreover authors should resume the main determinants of AF in their introduction (doi: 10.3390/medicina58111513.)
While most assertions are well-supported, the analysis leans heavily on a limited number of studies (e.g., FRAIL-AF). Including references to more recent and diverse research could enhance the strength of the argument.
The paper demonstrates good flow between sections, but care should be taken to ensure that each segment builds logically on the preceding one without unnecessary repetition.
Statements like "Frailty in AF patients represents a continuum of vulnerability rather than a binary state" are compelling but could be elaborated further to enhance clarity and depth.
The conclusion could be made more impactful by distilling key actionable insights for clinicians and researchers.
Author Response
Reviewer 2 comment:
- “The discussion on guideline implications would benefit from a more thorough comparison with current clinical practices. Moreover, authors should resume the main determinants of AF in their introduction (doi: 10.3390/medicina58111513).”
- “While most assertions are well-supported, the analysis leans heavily on a limited number of studies (e.g., FRAIL-AF). Including references to more recent and diverse research could enhance the strength of the argument.”
- “Statements like ‘Frailty in AF patients represents a continuum of vulnerability rather than a binary state’ are compelling but could be elaborated further to enhance clarity and depth.”
- “The conclusion could be made more impactful by distilling key actionable insights for clinicians and researchers.”
Author Response:
- Comparison with current clinical practices
We have revised the sections “Avoid Stroke and Thromboembolism” and “Evaluation and Dynamic Reassessment” to contrast the new ESC guidelines’ recommendation (maintaining VKA in stable older adults) with prevailing clinical practice, wherein many clinicians automatically shift patients to DOACs. We have also integrated the main determinants and mechanisms of AF by referencing and discussing findings from (doi: 10.3390/medicina58111513).
- Diverse research references beyond FRAIL-AF
We appreciate your note regarding reliance on the FRAIL-AF trial. We have added more data points and references to trials such as ARISTOTLE, ENGAGE AF-TIMI 48 and ELDERCARE-AF. These additions show a broader perspective on managing AF in older, often frail patients, thereby enriching our discussion with more recent and diverse evidence.
- Continuum of frailty
We have elaborated on the statement “Frailty in AF patients represents a continuum of vulnerability rather than a binary state,” clarifying how geriatric syndromes evolve over time.
More impactful conclusion with actionable insights
Our revised Conclusion specifically provides key takeaways for clinicians (e.g., the need for validated frailty assessments, careful DOAC selection, and multidisciplinary collaboration) and researchers (e.g., designing inclusive, pragmatic trials). We have reorganized the last paragraph to summarize high-yield recommendations that can be quickly translated into clinical practice.
Reviewer 3 Report
Comments and Suggestions for Authors
The authors have submitted a manuscript regarding the distinguish between frailty and disability among geriatric population and the presentation of AFib as a geriatric syndrome. A stronger clinical message has to be sent to the average reader in order to help the evaluation of patients. Other additional clinical examples according to guidelines have to be added. References are appropriate and the structure of the manuscript does not seem to need improvement.
Author Response
Reviewer 3 comment:
“The authors have submitted a manuscript regarding distinguishing between frailty and disability among the geriatric population and the presentation of AF as a geriatric syndrome. A stronger clinical message has to be sent to the average reader in order to help the evaluation of patients. Other additional clinical examples according to guidelines have to be added. References are appropriate and the structure of the manuscript does not seem to need improvement.”
Author response:
We have strengthened the clinical message throughout the manuscript by providing clearer, more direct guidance on patient evaluation and management. Specifically, in the “Comorbidity and Risk Factor Management” and “Reduce Symptoms by Rate and Rhythm Control” sections, we now offer short case-based examples demonstrating how a clinician might apply frailty screening, polypharmacy review, and individualized therapeutic strategies. In addition, we introduced new references that complement and update the existing guideline-based statements, ensuring our discussion is both up-to-date and clinically oriented.
Reviewer 4 Report
Comments and Suggestions for Authors
The authors assessed the association between AF and frailty or disability. They concluded emphasise the multifaceted needs of older adults through multidisciplinary collaboration and patient centered care.
I have the following concerns:
1. What kind of studies were included in the analysis?
2. What are the practical implications of the study?
3. Please define the terms: frailty nad polypharmacy
4. Which DOACs are preferred in population of elderly, frailty patients?
5. Please discuss symptoms management strategy in frailty patients.
Author Response
Reviewer 4 comment:
- “What kind of studies were included in the analysis?”
- “What are the practical implications of the study?”
- “Please define the terms: frailty and polypharmacy.”
- “Which DOACs are preferred in the population of elderly, frail patients?”
- “Please discuss symptom management strategy in frailty patients.”
Author Response:
- Nature of included studies
In our revised Introduction and “Evaluation and Dynamic Reassessment” sections, we clarify that our discussion synthesizes a range of studies (randomized controlled trials, observational cohorts, meta-analyses, and pragmatic trials). We highlight key evidence from FRAIL-AF, EUROSAF, ARISTOTLE, ENGAGE AF-TIMI 48, and ELDERCARE-AF as representative of contemporary data on older adults with AF.
- Practical implications
We have added a concise “Implications for Clinical Practice” subsection (within the Conclusion), summarizing the essential steps for clinicians—such as routine frailty assessments, thorough medication reviews, and interdisciplinary team involvement—to guide treatment decisions in older AF patients.
- Definitions: frailty and polypharmacy
In “Comorbidity and Risk Factor Management,” we have explicitly defined frailty as a state of increased vulnerability to stressors due to decline in physiological reserves, distinguishing it from disability (loss of independence). Similarly, we define polypharmacy as the concurrent use of five or more medications, emphasizing that its appropriateness hinges on whether each prescription aligns with the patient’s health goals, comorbidities, and life expectancy.
- Preferred DOACs in elderly, frail patients
In “Avoid Stroke and Thromboembolism,” we have expanded the discussion on how DOAC selection must consider renal function, bleeding risk, and patient compliance. We note that apixaban and edoxaban often demonstrate favorable safety profiles (less major bleeding, especially gastrointestinal) in older or frail patients. We also reference low-dose edoxaban data in very elderly cohorts from ELDERCARE-AF.
- Symptom management strategy in frail patients
We address symptom management in the “Reduce Symptoms by Rate and Rhythm Control” section, discussing both pharmacological (beta-blockers, calcium channel blockers, cautious digoxin use) and non-pharmacological measures (exercise programs, nutritional support, careful monitoring for delirium). We also added specific points on advanced therapeutic options like catheter ablation for appropriate candidates, while underscoring the need for regular, dynamic reassessment of frailty status.
Final remarks
We genuinely appreciate the comprehensive feedback from the reviewers. Their constructive critiques and suggestions have led us to refine our manuscript substantially, making it more clinically relevant and scientifically robust. We hope that our revisions address all concerns to your satisfaction and enhance the manuscript’s contribution to the field of geriatric cardiology.Thank you for your consideration, and we look forward to any further guidance you may provide.
Round 2
Reviewer 2 Report
Comments and Suggestions for Authors
I sincerely appreciate the effort and attention you have dedicated to addressing my comments during the revision of your manuscript. Your skill in integrating feedback while preserving the essence of your original vision highlights both your scientific proficiency and your dedication to delivering high-quality research.
Reviewer 4 Report
Comments and Suggestions for Authors
Thank you for your revision. I have no further concerns